# Ionospheric Plasma Flows Associated with the Formation of the Distorted Nightside End of A Transpolar Arc

Motoharu Nowada[1], Adrian Grocott[2], and Quan-Qi Shi[1]

[1] Shandong Key Laboratory of Optical Astronomy and Solar-Terrestrial Environment, School of Space Science and Physics, Institute of Space Sciences, Shandong University, Weihai, Shandong, 264209, People's Republic of China.

[2] Space and Planetary Physics Group, Department of Physics, Lancaster University, LA1 4YW, Lancaster, UK.

*Correspondence to*: Motoharu Nowada (moto.nowada@sdu.edu.cn)

**Abstract.** We investigate ionospheric flow patterns occurring on 28 January 2002 associated with the development of the nightside distorted end of a "J"-shaped Transpolar Arc (nightside distorted TPA). Based on the nightside ionospheric flows near to the TPA, detected by the SuperDARN radars, we discuss how the distortion of the nightside end toward the pre-midnight sector is produced. The "J"-shaped TPA was seen under southward Interplanetary Magnetic Field (IMF) conditions, in the presence of a dominant dawnward IMF-$B_y$ component. At the onset time of the nightside distorted TPA, particular equatorward plasma flows at the TPA growth point were observed in the post-midnight sector, flowing out of the polar cap and then turning toward the pre-midnight sector of the main auroral oval along the distorted nightside part of the TPA. We suggest that these plasma flows play a key role in causing the nightside distortion of the TPA. SuperDARN also found ionospheric flows typically associated with "Tail Reconnection during IMF Northward Non-substorm Intervals" (TRINNIs) on the nightside main auroral oval before and during the TPA interval, indicating that nightside magnetic reconnection is an integral process to the formation of the nightside distorted TPA. During the TPA growth, SuperDARN also detected anti-sunward flows across the open-closed field line boundary on the dayside that indicate the occurrence of low-latitude dayside reconnection and ongoing Dungey cycle driving. This suggests that nightside distorted TPA can grow even in Dungey-cycle-driven plasma flow patterns.

## 1. Introduction

Transpolar arcs (TPA) are the bar-shaped part of "theta aurora", connecting the nightside and dayside auroral ovals within the polar cap (Frank et al. 1982). Since theta auroras were discovered in the beginning of 1980's, TPAs have been the focus of much research, and various formation mechanisms have been proposed based on investigations of the ionospheric flow patterns and the relationship to the orientation of Interplanetary Magnetic Field (IMF) (see a series of reviews on polar cap arcs and TPAs by Hosokawa et al. 2020, Fear and Milan 2012a, Mailyan et al. 2015).

The TPA formation model based on nightside magnetic reconnection occurring under northward IMF conditions, which was proposed by Milan et al. (2005), has had a high degree of success in explaining a wide variety of TPA observations (e.g., Fear and Milan 2012a,b; Kullen et al. 2015; Nowada et al. 2018 and references therein). In this model, nightside magnetic reconnection forms closed magnetic field lines whose northern and southern footpoints straddle the midnight meridian. As a

result, the newly closed flux has no preferential return path to the dayside (i.e., via dawn or dusk) and instead protrudes into the magnetospheric lobe and thus into the polar cap ionosphere. This protruding closed flux becomes what we call the TPA, which in the simplest case grows straight from the nightside to the dayside in the polar cap. In the ionosphere, azimuthal plasma flows across the midnight meridian, normally ranging between about 300 m/s and 700 m/s but sometimes being faster than 700 m/s, are observed in the nightside auroral oval. These characteristic ionospheric flows are interpreted as evidence for nightside reconnection (e.g., Grocott et al. 2003, 2004), and are often referred to as the flow signatures associated with "Tail Reconnection during IMF Northward Non-substorm Intervals (TRINNIs)" (e.g., Milan et al. 2005). Such flows are observed at the poleward edge of the main nightside auroral oval, which is the boundary between open and closed magnetic flux, in the vicinity of the growth point of TPA. This indicates that magnetotail magnetic reconnection occurs close to the region of the TPA formation (Milan et al., 2005; Fear and Milan 2012b). However, we might also expect to find a region of much slower flow at the site of the TPA growth itself, since the TPA formation mechanism is directly related to a stagnation of the newly closed magnetic flux and associated plasma flows.

TPAs are sometimes seen during southward IMF intervals. However, in most of those cases, there had been prolonged northward IMF intervals before the TPA occurrences (e.g., Craven et al. 1991, Newell and Meng, 1995, Pulkinnen et al. 2020, and references therein). Certainly, in particular southward IMF TPA cases discussed by Craven et al. (1991) and Newell and Meng, (1995), the magnetospheric and ionospheric dynamics triggered by the change in IMF orientation from northward to southward seemed not to have played an essential role in the TPA formation processes. However, using on a combination of auroral imager observations and MHD global simulations of magnetotail dynamics, Pulkkinen et al. (2020) recently suggested that fast plasma flows triggered by strong magnetotail reconnection in the distant magnetotail may be a "source" of TPAs under southward IMF conditions.

In contrast to these straightforward TPAs (hereafter, referred to as "regular TPA") frequently occurring under northward IMF conditions, bending or "hooked-shaped" arcs have also been reported. These arcs grow from the dawnside or duskside main auroral oval to the dayside. They are observed when the IMF orientation is southward, or when it turns from long-term southward (northward) to northward (southward) in the presence of a dominant IMF-$B_y$ component (Kullen et al. 2002, 2015; Carter et al. 2015). Their formation can be explained by magnetic reconnection at the low-latitude dayside magnetopause (Kullen et al. 2015, Carter et al. 2015). Carter et al. (2015) proposed the detailed formation process of bending arcs, which are formed by the entry of the solar wind (magnetosheath) particles along open field lines generated by low-latitude dayside reconnection. Their growth toward pre- or post-noon is caused by dawn-dusk asymmetric ionospheric plasma convection caused by the presence of IMF-$B_y$ penetration (e.g., Cowley and Lockwood, 1992). TRINNI flows were not found during the bending arc development, suggesting that nightside magnetic reconnection is not related with the formation of bending arcs. (Kullen et al., 2015, Carter et al. 2015).

Nightside distorted TPAs are duskside (dawnside) TPAs with their nightside ends distorted toward post- (pre-) midnight, and were first identified based on a statistical study by Fear and Milan (2012b). Nowada et al. (2020) proposed a possible formation scenario of the nightside distorted TPAs. According to their scenario, the essential source of nightside distorted

TPAs is upward (flowing out of the ionosphere to the magnetotail) field-aligned currents (FACs), which are generated by plasma flow shear between fast plasma flows triggered by magnetotail magnetic reconnection and slower background magnetospheric flows. They also postulated that the TPA growth to the dayside is attributed to retreat of the magnetotail reconnection points to further down-magnetotail. During the development of nightside distorted TPAs, as the reconnection site goes further tailward, the magnetotail gets more deformed and associated field lines are also twisted more strongly (Tsyganenko et al. 2015; Tsyganenko and Fairfield, 2004), caused by the IMF-$B_y$ penetration (Gosling et al. 1990; Cowley, 1981, 1994). Nowada et al. (2020) concluded that owing to the magnetotail deformation and field line twisting, the TPA does not straightforwardly grow from the nightside main auroral oval to the opposite dayside oval, but develops with a "distortion" of its nightside end toward dawn or dusk. However, in-situ observational evidence on the TPA deformation is yet to be detected in either the magnetosphere or ionosphere.

The nightside distorted part of the TPA is frequently aligned with main auroral oval, but is a distinct feature at the auroral oval's poleward edge. This might be related to other observations where a part of the nightside auroral oval appears as bifurcated branches, equatorward and poleward, with a gap (or a weak emission region) between them. This separated auroral feature is identified as a "double auroral oval" (e.g., Elphinstone et al., 1995a,b). Ohtani et al (2012) investigated the detailed electric current structures and formation mechanism of a double auroral oval seen in the dusk-midnight sector. Such double auroral ovals are frequently seen under geomagnetically active conditions, and during the latter (recovery) phase of intense polar substorms. The equatorward branch of the double auroral oval is embedded in upward field-aligned currents (FACs), flowing out of the ionosphere to the magnetotail, with downward FACs from the magnetotail to the ionosphere dominantly collocated in the double oval poleward branch. Each branch of the bifurcated auroral oval therefore connects to different region of the magnetotail. Ohtani et al. (2012) concluded that the equatorward branch connected the field lines at geosynchronous altitudes and outside, corresponding to the ring current and the near-Earth part of the tail current, while the poleward double oval branch mapped to a wide area farther down-tail, where accelerated auroral particles precipitating to the poleward branch are generated.

In this paper, we report a significant finding in relation to the ionospheric plasma flows that may explain the generation of the nightside distorted end of the TPA. This study is achieved using ionospheric flow patterns measured by the SuperDARN (Super Dual Auroral Radar Network) HF (High Frequency) radars and auroral imager data obtained from the Imager for Magnetopause-to-Aurora Global Exploration (IMAGE) satellite.

This paper consists of six sections. The introduction and the instrumentation used in this study are given in sections 1 and 2. Introduction of the nightside distorted TPA is shown in section 3. In section 4, the observational results of solar wind conditions, global ionospheric plasma flows associated with the formation of nightside distorted TPA are reported. Finally, in sections 5 and 6, we present discussions and our conclusions of this study.

## 2. Instrumentation and Data Processing

### 2.1 Auroral Images

Nightside distorted TPAs were identified using auroral observations by the Wideband Imaging Camera (WIC), which is part of the Far Ultraviolet (FUV) instrument (Mende et al., 2000a, 2000b, 2000c) onboard the Imager for Magnetopause-to-Aurora Global Exploration (IMAGE), launched in March, 2000. IMAGE FUV-WIC imaged the aurora in a broad wavelength range from 140 nm to 190 nm with 2 minutes cadence. The IMAGE FUV-WIC data includes non-auroral optical signals due to sunlight (dayglow) and the instrumental optical noise. In this study, we have removed non-auroral data as much as possible using the methods described in Nowada et al. (2020).

### 2.2 Ionospheric Convection Maps

Ionospheric plasma flow data were obtained by the SuperDARN HF radars (Greenwald et al., 1995; Chisham et al., 2007). These high-latitude radar arrays in both northern and southern hemispheres make line-of-sight measurements of ionospheric flow velocity. For this study, data from all radars in the northern hemisphere were combined using the 'map potential' technique (Ruohoniemi and Baker, 1998) which fits an $8^{th}$ order spherical harmonic expansion of the ionospheric electric potential to the measured flows to provide large-scale maps of the ionospheric convection pattern. This is achieved by first median averaging the line-of-sight data onto an equal area magnetic latitude and longitude grid, within cell size $\sim 110 \times 110$ km, to remove anomalous data. A lower-latitude boundary to the convection is then estimated from the distribution of measured velocities as the lowest latitude at which a threshold of at least 3 measurements of $\sim 100$ m/s is met. The background statistical model of Ruohoniemi and Greenwald (1996) is then used to provide a set of vectors that supplement the observations to provide enough measurements for the spherical harmonic fit to converge.

As recently discussed by Walach et al. (2022), the map potential solutions are sensitive to a number of factors that govern the resulting convection maps. One factor is the latitudinal extent of the radar coverage. For example, more recent additions to the radar network at mid-latitudes can improve estimates of the flow in these regions. However, in this study, only the auroral zone SuperDARN radars had been built. This is unlikely to affect our results owing to the relatively contracted polar cap in this case; the flows of interest were located at ~70° magnetic latitude and thus well within the fields-of-view (FOV) of the auroral zone radars. Another factor is the placement of the equatorward boundary of the convection – the so-called Hepner-Maynard boundary (HMB; Heppner and Maynard, 1987). This can be influenced by irregular data coverage and also by the inclusion of slower E-region scatter which can be found at near-ranges in the radar FOV. We therefore carefully inspected the placement of the HMB in our analysis and modified the automatically generated boundary to remove unphysical steps that occasionally occurred due to the inclusion of possible E-region contamination (near-range, slow flows). We further compared the boundary to the auroral images and found that it was generally located close to the equatorward edge of the auroral oval, as expected. Finally, we also considered the choice of background model, in which Walach et al. (2022) found to influence the map potential output where the number of measurements is low. We decided upon the Ruohoniemi and Greenwald (1996) model over more recent options for two main reasons. First, this model was derived from

auroral zone data for only a few years prior to our interval. This might be expected to make it more appropriate than more recent models, such as Thomas and Shepherd (2018), constructed from data based on a different solar cycle, and including data from different geophysical regions. Second, the Ruohoniemi and Greenwald model explicitly focuses our interpretation on regions in which direct measurements exist, such that the choice of model has anyway less influence on the resulting flow vectors.

All SuperDARN convection maps shown in this study were produced based on the aforementioned processing steps. Each map includes streamlines of the electric equipotentials and their values with black solid and broken contours on the dusk and dawn sides, respectively. Convention is to assign the opposite signs to the ionospheric electrostatic potential with positive and negative vorticity; the electric potential at dawn is generally positive (maximum potential denoted by a plus sign), and at dusk is generally negative (minimum shown by a cross). To estimate the two-dimensional flow velocities, we use the available radar line-of-sight measurements, with the transverse velocity component derived from the map potential solution. We choose to present these composite vectors, rather than the commonly used global electrostatic solutions, because we are mainly interested in the flows driven by dynamic magnetotail processes that are not well resolved in the global patterns.

### 3. Overview of Nightside Distorted TPAs

The first (left) panel in Figure 1 shows an example of a "regular" TPA with straight bar-shaped emissions, connecting the nightside and dayside auroral ovals. In nightside distorted TPAs, as shown in the second and third panels, their nightside ends get distorted toward the pre- or post-midnight sectors, respectively. Nowada et al., (2020) identified the TPAs with the nightside distortions as "J"- and "L"-shaped TPAs based on their resemblance to the letters "J" and "L". Taking a look at the "J" and "L"-shaped TPAs, the distorted directions of their nightside ends are, of course, opposite to each other, but otherwise no significant difference can be seen, particularly, in the emissive parts that straightforwardly cross the polar cap to the dayside. Most cases of nightside distorted TPA were observed during northward IMF intervals with a dominant IMF-$B_y$ component (Nowada et al. 2020). However, some "regular" and nightside distorted TPAs can be observed even under southward IMF, but usually where the IMF orientation had been previously persistently northward (Fear and Milan (2012a,b or see the nightside distorted TPA event list in Nowada et al. 2020).

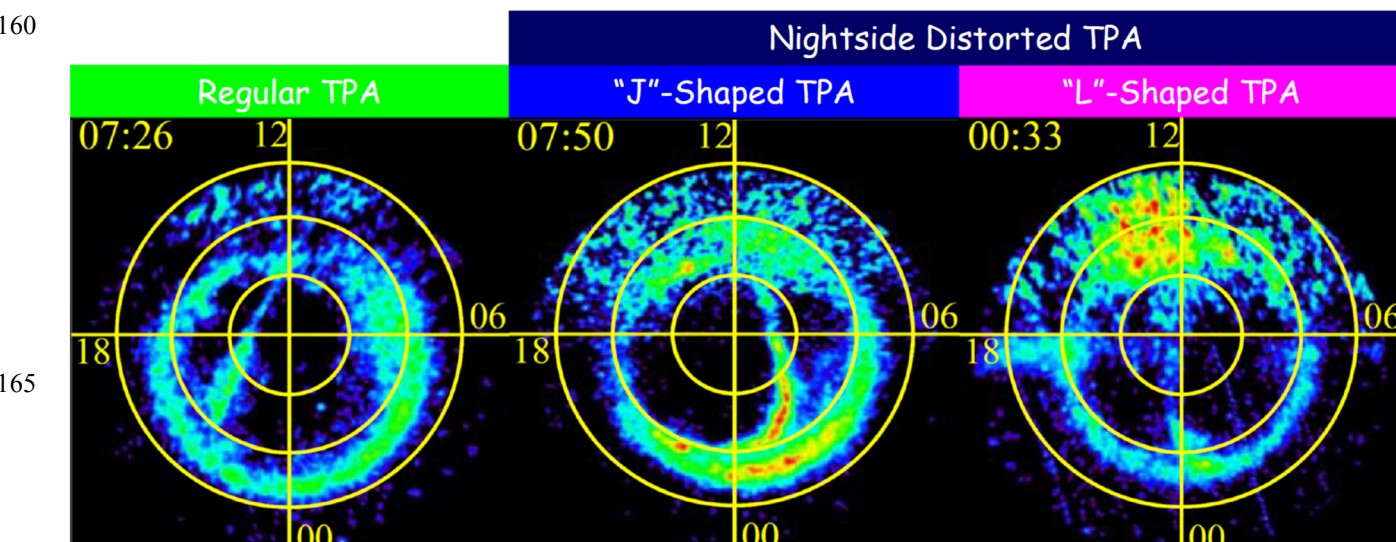

Figure 1: Representative examples of the IMAGE FUV-WIC observations for three types of the transpolar arc (TPA) morphologies; "Regular" TPA (left) and nightside distorted TPAs, including "J"- (center) and "L"- (left) shaped TPAs, as observed on 5 November , 2000, 22 September 22, 2000, and 12 March, 2002 are shown, respectively. These nightside distorted TPAs are referred from the nightside distorted TPA database, provided from Nowada et al. (2020). The top and bottom sides in each panel show the noon and midnight sides whose magnetic local times (MLTs) are 12h and 24h, respectively. The dawn (6h MLT) and dusk (18h MLT) meridians correspond to the right and left sides, respectively. The concentric circles in each panel show the magnetic latitude (MLat) at 60, 70, and 80 degrees, respectively. The color code is expressed in Analogic-Digital Units (ADU), which is proportional to the observed auroral brightness (see the details in Mende et al. 2000b). The dayglow and background optical noises from the IMAGE FUV-WIC data are removed by the techniques described in Nowada et al. (2020).

## 4. Observations of Solar Wind Conditions, Ionospheric Flows Associated with the Nightside Distorted TPA

### 4.1 Formation of Closed Magnetic Field within the "J"-shaped TPA

In Figure 2 shows a case of a nightside distorted TPA ("J"-shaped TPA) which was observed on 28 January, 2002, along with corresponding geomagnetic activity, IMF conditions and global ionospheric flows. We identified the onset time (9:25:27 UT) of this distorted TPA based on the IMAGE FUV-WIC data via visual inspection. In Figure 2(a), the geomagnetic activity, and IMF conditions obtained from the OMNI solar wind database during a 4 hour-interval between 7:30 UT and 11:30 UT are shown. The time interval corresponding to the nightside distorted TPA observation is bracketed

by two gold broken lines. The black dotted lines with the labels from (a) to (h) indicate 8 key times of interest discussed later and shown in Figure 3. During the interval of the nightside distorted TPA, the magnetosphere was geomagnetically quiet, with $AL$ and $AU$ index values smaller than -20 nT and 35 nT, respectively. The associated IMF-$B_y$ component was dominantly negative (dawnward).

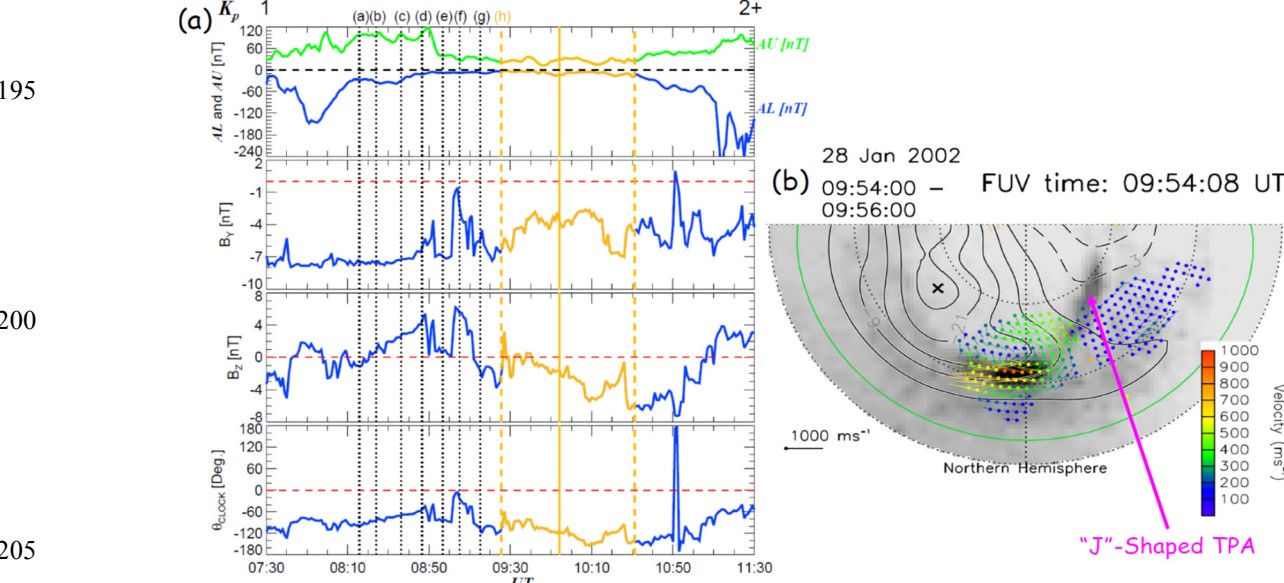

**Figure 2: Panel (a) shows the plots of the geomagnetic activity indices and OMNI solar wind data during 4 hours between 7:30 UT and 11:30 UT on 28 January, 2002 are displayed; the $AL$ and $AU$ indices, representing the geomagnetic activity at high-latitudes, the Y (dawn-dusk) and Z (north-south) components of the IMF in GSM coordinates, and the IMF clock angle, calculated with a formula of arctan(IMF-$B_y$/IMF-$B_z$). The $K_p$ index to represent the average global geomagnetic condition during the interval of interest is also shown onto the $AL$ and $AU$ plot panel. The TPA interval is bracketed by two gold broken lines and the solar wind condition corresponding to 9:54:08 UT (panel b) is marked by a gold sold vertical line. The black dotted lines with the labels from (a) to (h) on the top panel show the solar wind conditions corresponding to the times of the 8 IMAGE FUV-WIC/SuperDARN plots in Figure 3. Panel (b) shows the nightside ionospheric flow velocity vectors measured by SuperDARN, overlaid onto the snapshot of the IMAGE FUV-WIC data, including the aurora oval and nightside distorted ("J"-shaped) TPA seen on 9:54:08 UT in the northern hemisphere in geomagnetic coordinates. The concentric dotted circles show the magnetic latitude (MLat) at 60, 70, and 80 degrees, respectively. The left, bottom and right sides in each panel show 18h, 24h and 6h in magnetic local time (MLT), respectively. The green curves show the Heppner-Maynard boundary, which is the lower latitude limit for the ionospheric plasma convection pattern. The vector length and color code are assigned according to the flow orientation and intensity of ionospheric plasma velocity in units of m/s.**

The dip in the *AL* index down to -150 nT between 7:30 UT and 8:10 UT suggests that an auroral substorm occurred more than 1 hour prior to the TPA onset time (9:25 UT). The AL magnitude subsequently decreased, that is, the substorm entered the recovery phase from 8:10 UT to 8:50 UT. A larger substorm, whose *AL* peak was over -240 nT, occurred after the disappearance of the TPA (10:31 UT). Therefore, the "J"-shaped TPA was seen during a geomagnetically quiet interval in the polar region between two auroral substorms. Such quiet magnetospheric conditions are favorable for the formation of nightside distorted TPAs.

During the interval from at least 1 hour prior to the onset of the arc (~ 8:20 UT ~ 9:20 UT), the IMF-$B_z$ component was predominantly northward. However, from 9:20 UT, the IMF-$B_z$ component turned to, and persisted in, a southward orientation. The "J"-shaped TPA was seen under almost entirely southward IMF conditions, although the IMF-$B_z$ component transiently turned northward just after the onset of the nightside distorted TPA, and fluctuated between north- and southward directions until ~9:42 UT. After these fluctuations, the orientation of the IMF-$B_z$ component persisted southward. During the TPA interval, the average clock angle was ~ -120 degrees, due to a strong IMF-$B_y$ component. The solar wind plasma conditions were stable during the TPA interval (not shown here).

Figure 2(b) shows an example of the plasma flow velocity vectors between 9:54 UT and 9:56 UT from the nightside ionosphere covering the magnetic local time (MLT) range from 18h to 6h, overlaid on a greyscale image of main auroral oval and "J"-shaped TPA detected by IMAGE FUV-WIC in the Northern Hemisphere at 9:54:08 UT. The time of this image corresponds to the gold solid vertical line on the solar wind data in panel (a). The "J"-shaped (nightside distorted) TPA is comprised of the "bar part", which is growing toward the dayside with a slight dawnward sense in the post-midnight sector and the nightside end, distorted toward the pre-midnight sector. The associated velocity vectors are projected onto a geomagnetic grid. The green curve shows the HMB, that is, the lower latitude limit for the ionospheric plasma convection pattern. The vector length and color code are assigned according to the intensity of ionospheric flow velocity in units of m/s. Around the poleward edge of the main auroral oval, westward flows from the post-midnight to the pre-midnight sector whose speed is between 400 m/s and 750 m/s were observed. These ionospheric flow signatures can be seen to have crossed from poleward of the main aurora oval just post-midnight, and adjacent to the TPA growth point. Assuming that the poleward edge of the auroral oval is a proxy for the boundary between the open and closed magnetic flux regimes, then these flows would thus appear to be associated with magnetotail reconnection. Although the flows here occurred during an interval of southward IMF, they have the same significant characteristics as the ionospheric flow signatures of Tail Reconnection during IMF Northward Non-substorm Intervals (TRINNI) flows (e.g., Grocott et al. 2003, 2004); in this case westward fast plasma flows at the poleward edge of the main auroral oval across the midnight meridian. According to the average statistical picture of the ionospheric return flow, given by Reistad et al. (2018), TRINNI-type return flows can be seen even under the southward IMF as long as the IMF-$B_y$ component is present (as is the case shown here). Furthermore, the TRINNIs occur within global ionospheric convection flow, excited by ongoing and modest dayside reconnection, and under an absence of substorm activity, which are all true of the present interval. The presence of TRINNI-type flows explicitly suggests that nightside magnetic reconnection has occurred, and thus plays a role in the formation of this "J"-shaped TPA,

providing a source of closed flux, as previously proposed by Nowada et al. (2020). Considering that the dynamic properties of a TPA within the polar cap, such as its drift motion, are governed by global ionospheric flows (c.f. Milan et al., 2005, Fear et al. 2015), our observation of plasma flows near the nightside end of a TPA may be the key phenomenon to understanding the cause of the distortion. The ionospheric flow velocity on the nightside distorted part of the TPA was faster than that on the poleward edge of the main auroral oval. Therefore, the nightside distorted part appears to be distinct from the main auroral oval in this case.

Figure 3 shows a time series of the overlaid plots of IMAGE FUV-WIC and SuperDARN radar data from 70 minutes prior to the onset time of the "J"-shaped TPA at 9:25:27UT. A number of features can be seen in the nightside flow during this time. In panels (a) and (b), 70 to 60 minutes prior to the "J"-shaped TPA onset time, any significant flows seem to be restricted to the pre-midnight sector, associated with structure in the brighter parts of the pre-midnight auroral oval. These intensifications of the poleward boundary, that is, Poleward Boundary Intensification (PBI), are likely to be features of the recovery phase of the preceding substorm (e.g., Lyons et al., 1999). The flows in the post-midnight have become enhanced by panel (c), about 50 minutes prior to the TPA onset. These flows seem to cross the polar cap boundary at around $1 \sim 2$ MLT. In panel (d), about 40 minutes prior to onset, the flows can be seen to have moved to earlier local times, in concert with the PBI also having moved to earlier MLT ($\sim 0$ MLT). Now, flows across the poleward edge of the main auroral oval around the midnight sector can be seen; these are highlighted by the zoomed-up flow profiles in the orange-framed region. These flow signatures were observed until the onset of the TPA, as also shown in the orange-framed magnifications of panels (e) – (h). From 30 minutes prior to the TPA onset (panels e – h), these flows evolve into the classical signature of TRINNIs – oval-aligned return flows – that act to remove and redistribute the closed flux into the auroral zone at earlier local times.

The ongoing observation of TRINNIs from -30 minutes up to the onset time of the TPA suggests that nightside reconnection persisted around the TPA growth point. However, of particular significance is that, at the times shown in panels (g) and (h) of Figure 3, there also exists a region dawnward of the TRINNI return flows that consists of slow (and stagnant) flows lower than 250 m/s, indicated with blue-colored vectors. Subsequently, the TPA began to grow within these stagnant flow regions, which is consistent with the conventional TPA formation model based on nightside magnetic reconnection, as proposed by Milan et al. (2005). Indeed, this stagnant flow region can still be seen in the flow pattern at 9:47:59 UT in Figure 5(b), when the TPA was still growing toward the dayside. Based on these TRINNI flow profiles detected by the SuperDARN radar measurements, we can infer that nightside reconnection occurred before and at the TPA onset time, and is thus likely to be part of the mechanism for the formation of the "J"-shaped TPA. This is consistent with the fundamental formation scenario of the nightside distorted TPA, proposed by Nowada et al. (2020). It still remains unclear, however, how the nightside distortion of the TPA was formed.

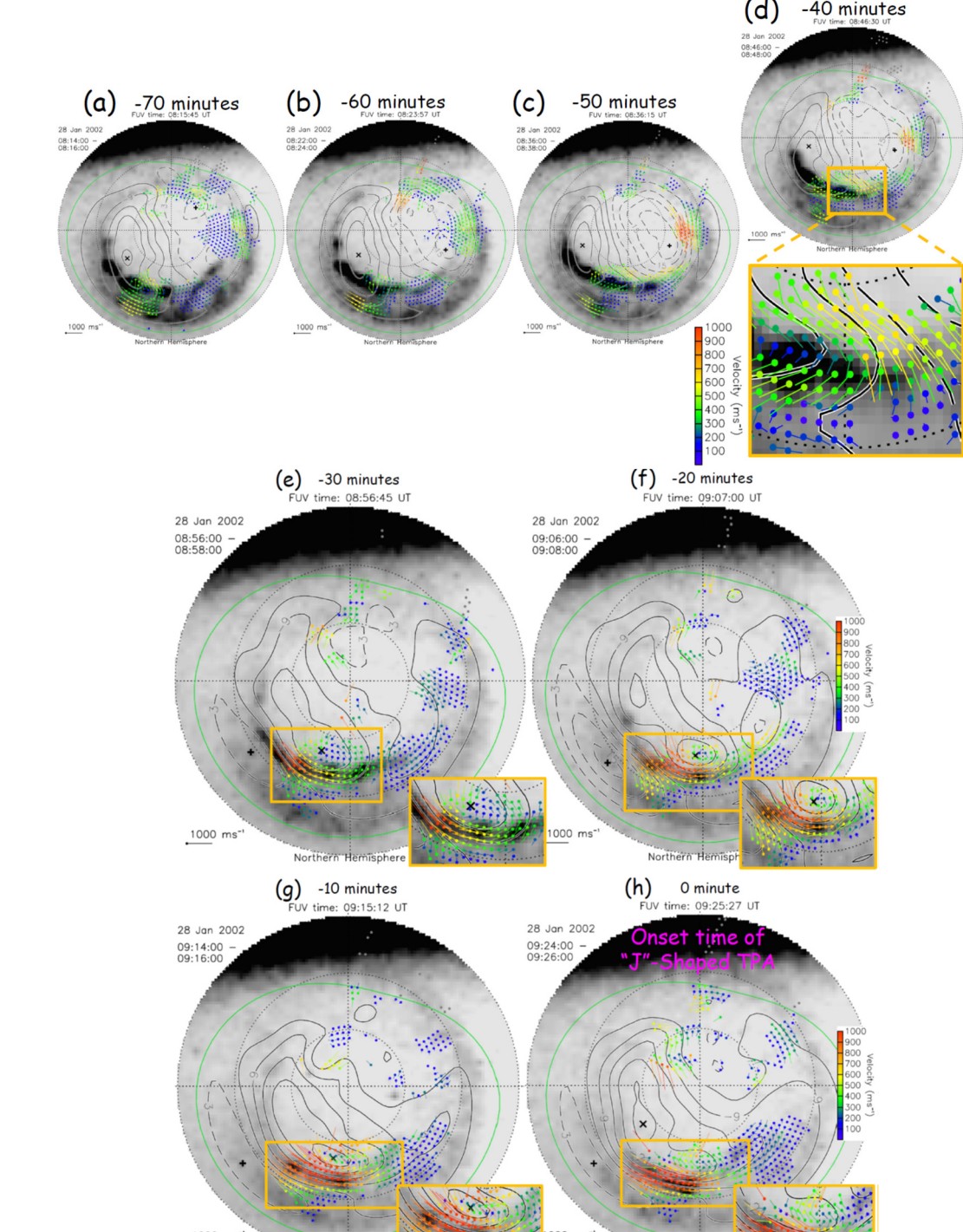

**Figure 3: The eight selected overlaid plots of the SuperDARN radar and IMAGE FUV-WIC data are displayed.**
**From panels (a) to (g), the overlaid plots between 70 minutes and 10 minutes prior to the onset time of "J"-shaped**
**TPA (9:25:27 UT), displayed in panel (h), are shown. Zoomed-up ionospheric TRINNI flow profiles in the regions**
**including the midnight sector are shown within orange-framed boxes. The dayside auroral images and SuperDARN**
**data are included, although the plot format in panels shown is basically the same as panel (b) in Figure 2.**

We consider that the TRINNI return flows themselves may be involved in the formation of the nightside end distortion of a TPA. In Figure 4(a) which shows an overlaid plot of the IMAGE FUV-WIC and SuperDARN radar data at 8:23:57 UT, we find no significant TRINNI return flows across the post-midnight sector of main auroral oval, suggesting that nightside reconnection was not occurring at this location, where the TPA subsequently formed. However, there were weak plasma flows which crossed the open/closed field line boundary, which is inferred to be collocated with the poleward edge of the auroral oval, in the pre-midnight sector. In Figure 4(b) (9:15:12 UT), prior to the subsequent "J"-shaped TPA onset (9:25:27 UT, Figure 4c), fast ionospheric TRINNI-type flows (~ 1,000 m/s) were seen in the midnight sector. These flows explicitly support the idea that nightside reconnection was occurring at this time. Furthermore, the flows were rotating their orientation toward the west (duskside) across the open/closed polar cap boundary, thus also providing evidence for the return of newly closed magnetic flux toward the dayside. At the "J"-shaped onset time in Figure 4(c), the TRINNI flows were still observed around the midnight sector, approximately 1h MLT earlier than the TPA growth location. This indicates that nightside reconnection occurred across the midnight sector at the TPA onset time, but not at the precise location of the TPA growth. The presence of the extremely slow flows (blue-colored vectors) around the growth point of the TPA, however, suggests that newly closed flux here is not "returning", but "stagnating". This is consistent with the idea that any flux closed in this sector would contribute to form the TPA.

This transition from equatorward flow across a stationary reconnection line (the "regular" pre-TPA case), to stagnant flows and a poleward-moving reconnection line (TPA growth case; see Figure 7 in Nowada et al. 2020), can be seen when comparing the flows around the location where the TPA protrudes into the polar cap between panels (b) and (c). Of particular interest is that the equatorward plasma flow region has not ceased but has clearly moved from the TPA location (post-midnight) to being adjacent to the base of the TPA in the midnight sector. This implies that the stagnant flux of the TPA is azimuthally restricted, and that newly closed flux adjacent to the TPA is still contributing to a TRINNI return flow channel. Furthermore, plasma flows at the eastward end of the nightside distorted part of the TPA (i.e. the part adjacent to the main oval) also appear to be flowing out of the TPA and along the main auroral oval. This suggests that the field lines mapping to the equatorward end of the TPA, which corresponds to the closed lobe flux crossing the equatorial plane nearest to the Earth, have started to flow out of the TPA where they also contribute to the TRINNI-type return flow channel. We suggest that the nightside end of a TPA is distorted by these duskward plasma flows, which are flowing along the distorted nightside end of the TPA. The presence of two potential sources of return flux – the nightside end of the TPA and the TPA-

adjacent flows – may explain the presence of the distinct auroral feature that forms the base of the "J"-part of the TPA adjacent to the main oval.

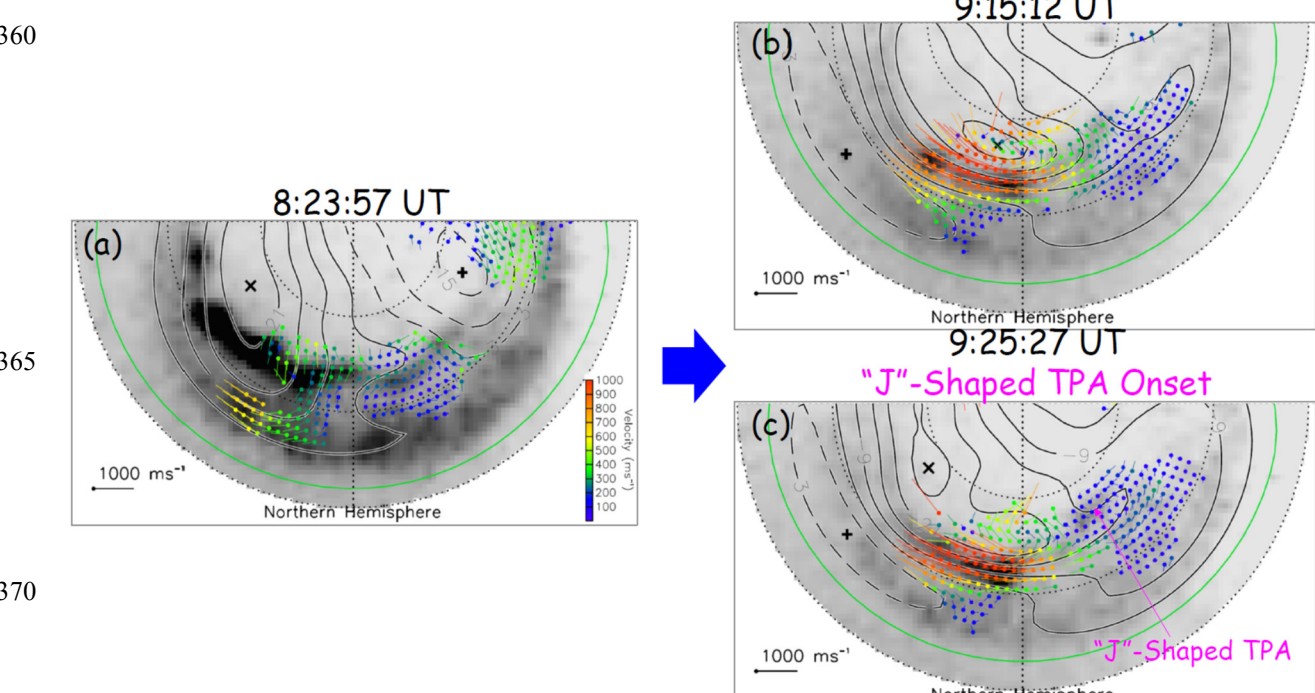

**Figure 4:** The three selected snapshots of the overlaid SuperDARN radar and IMAGE FUV-WIC data to investigate the TRINNI flows and polar cap convection patterns are shown. These overlaid plots are zoomed-up to the nightside sector from 18h to 6h in MLT. The overlaid plots about 1 hour, and at 10 minutes prior to the onset of the "J"-shaped TPA (9:25:27 UT) are shown in panels (a) and (b), respectively. Panel (c) displays the plot at the onset time of the "J"-shaped TPA. All figure formats are the same as those in Figures 2(b) and 3.

**4.2 Global Ionospheric Plasma Flows Driven by Dayside and Nightside Magnetic Reconnection**

Dayside magnetic reconnection globally drives the plasma flows within the polar cap (e.g., Dungey, 1961; Cowley and Lookwood, 1992). The TRINNI mechanism for generation of the flows at the distorted nightside end of the TPA requires ongoing dayside reconnection, although Fear et al. (2015) suggest that TPAs are associated with a suppression of open flux production at the dayside. Here, we briefly consider the global convection during the TPA interval. At the "J"-shaped TPA onset time (Figure 5a), the TPA just started to grow into the polar cap from the post-midnight auroral oval, marked with a yellow star. As discussed above, fast ionospheric TRINNI flows with a velocity of ~ 1,000 m/s were seen adjacent to this, providing clear evidence of nightside reconnection occurrence. During the growth of the nightside distorted TPA (Figure 5b), the ionospheric TRINNI return flows were still found in the vicinity of the poleward edge of the main auroral oval across the

midnight sector (particularly see the zoomed-up flow profiles in the orange-framed boxes). These flows suggest that magnetotail reconnection persisted even during the growth of the TPA, and formed the closed field lines of the distorted-part of the TPA. Further ionospheric flows along the distorted TPA nightside end were also observed.

Turning now to the dayside region, there were anti-sunward, and anti-sunward/duskward plasma flows which entered the polar cap across the open/closed field boundary (poleward edge of the dayside main auroral oval), highlighted by cyan boxes in Figure 5(b). These flow signatures provide key evidence for the occurrence of dayside reconnection (e.g., Cowley and Lookwood, 1992, Neudegg et al., 2000, Milan et al., 2000, and references therein). At the onset time of the "J"-shaped TPA (Figure 5a), dayside reconnection may have not yet been occurring because anti-sunward plasma flows across the dayside open/closed field line boundary were absent at this time, despite the IMF being oriented weakly southward (refer to interval (h) in Figure 2a). However, the ionospheric flows did subsequently begin to enter the dayside polar cap, and continued to do so while the nightside distorted TPA was growing from the nightside main auroral oval at 9:47:59 UT, completely reaching the dayside oval by 10:28:58 UT, as shown in two panels in Figure 5(b). It may be that ongoing dayside magnetic reconnection, and subsequent excitation of the TRINNI flows are required elements of the mechanism by which the nightside distorted ("J"-shaped) TPAs are produced.

When considering global ionospheric convection patterns during an interval of dominant dawnward IMF-$B_y$, we expect duskward flows in the dayside polar cap and dawnward flows in the nightside polar cap (e.g., Cowley and Lookwood, 1992). In this case, with a TPA growing in the post-midnight sector, dawnward plasma flows will thus be expected on the duskside of the TPA. The SuperDARN radars detected some evidence of these flows, just poleward of $\sim$ 80 degrees in MLat., as indicated by red boxes in Figure 5(b). Although the radar scatter within the polar cap is limited, the indicative flow pattern, seen in the two panels in Figure 5(b), is consistent with Dungey-cycle driving during a period of dawnward IMF-$B_y$ component under the southward IMF conditions in the northern hemisphere. The observed development of the "J"-shaped TPA from the nightside main auroral oval to the dayside was clearly not impeded by these Dungey-cycle-driven flows.

## 5. Discussions

### 5.1 Global Ionospheric Flow Patterns Associated with the "J"-shaped TPA Growth

In this paper, we have tried to unravel the formation of the nightside distorted part of the "J"-shaped TPA using ionospheric flow observations by the SuperDARN radars. When the nightside distorted TPA was observed, the ionospheric TRINNI "return" flows were seen on the main auroral oval across the midnight sector, suggesting a formation mechanism associated with nightside reconnection (e.g., Nowada et al. 2020). Interestingly, TRINNI flows were also observed in the region of the TPA growth, albeit at a reduced rate ($\sim$ 400 m/s) relative to the adjacent region (700 m/s $\sim$ 900 m/s) (see Figure 4). This suggests that the flows at the TPA base did not fully stagnate, but that their reduced rate might explain the build-up of closed flux. Their presence might also explain the nightside distorted part of the TPA, maintained by the newly closed flux being returned in the dusk convection cell by the TRINNI flow. This is consistent with the idea that the reduced-rate return flows are related to the nightside distortion of the TPA.

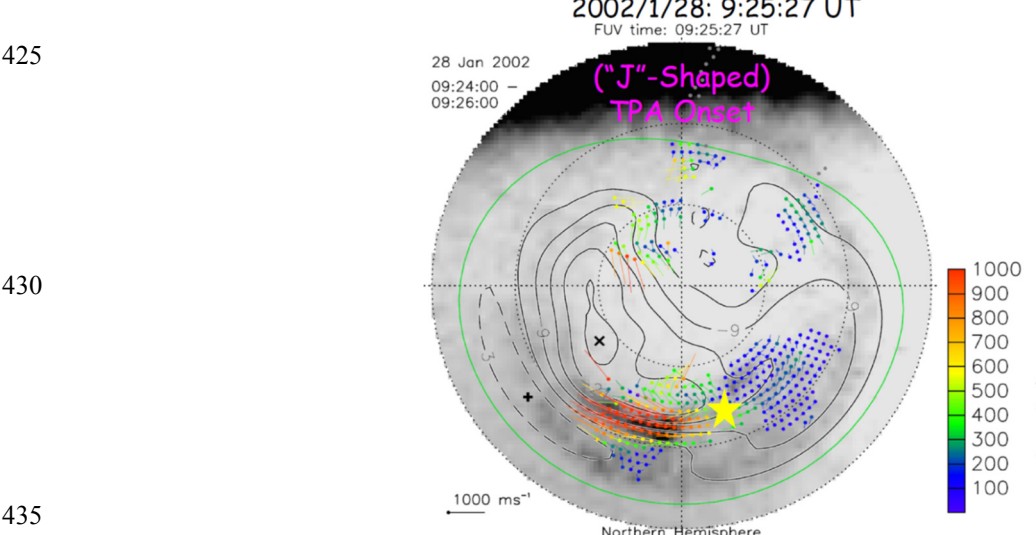

**Figure 5: (a) an overlaid plot of the ionospheric plasma flow pattern and IMAGE FUV-WIC imager data at the onset time (9:25:27 UT) of the "J"-shaped TPA is shown. The TPA growth point toward dayside is marked with a yellow star. (b) two panels show the overlaid plots on 9:47:59 UT and 10:28:58 UT when the fast flows entered the dayside polar cap across the poleward edge of the dayside main auroral oval, which is a proxy of open/closed field line**

**boundary, during the TPA growth, highlighted by cyan boxes. The dawnward ionospheric flows, which are consistent with the flow patterns as expected from Dungey-cycle driving during a period of dawnward IMF-$B_y$ component under the southward IMF (IMF-$B_z$ < 0), are surrounded by red boxes. The profiles of TRINNI-type westward fast flows around the midnight sector (highlighted with by orange boxes) are shown within the orange-framed box. All figure formats are the same as Figure 3.**

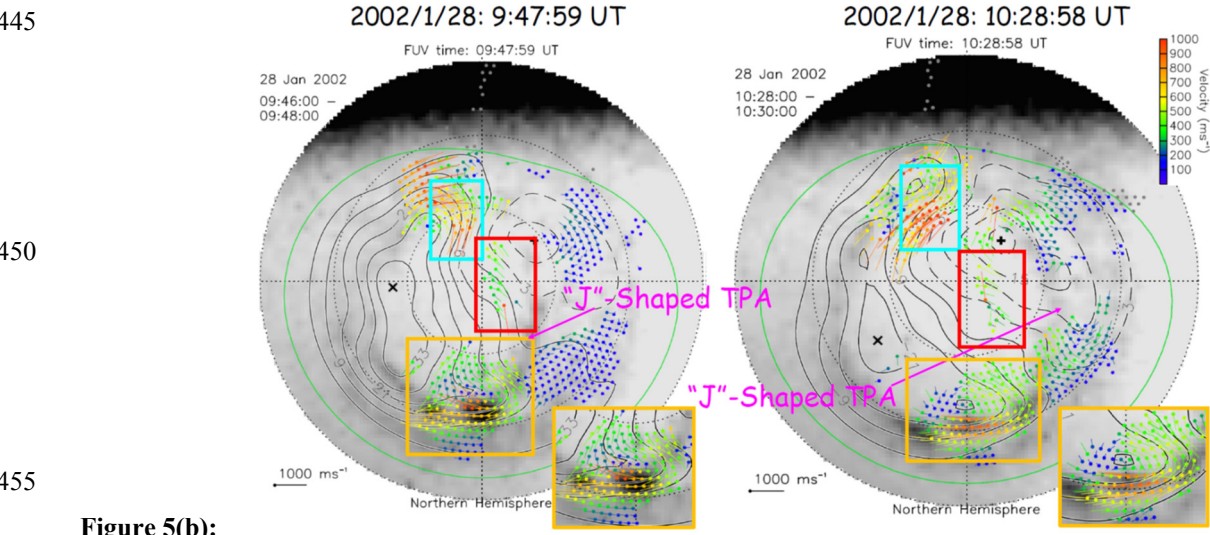

**Figure 5(b):**

Considering the global ionospheric flow patterns which are estimated based on the SuperDARN radar observations, and illustrated in Figure 6, the dayside reconnection-driven flows enter the dayside polar cap in the dusk sector, highlighted with thick cyan curved arrows and box, and are consistent with the plasma flows shown in cyan boxes in Figures 5(b). Furthermore, these flows subsequently feed the dawnward flows at higher latitudes, as highlighted with thick red curved arrows in the red box, which are being primarily driven by nightside magnetic reconnection. This is then also consistent with the post-midnight origin of the nightside reconnection flows close to the growth point of the TPA. However, it remains unclear whether or not the TRINNIs originate at the same down-tail location as the closed field lines of the "J"-shaped TPA. Despite the persistence of Dungey-cycle plasma flow patterns being driven by ongoing low-latitude dayside reconnection, which might be expected to inhibit the growth of a TPA, the "J"-shaped TPA observed here ultimately grew across the polar cap to the dayside. The "J"-shaped TPA's formation process is consistent with the model proposed by Nowada et al. (2020), that is, closed field lines associated with the TPA are formed by nightside magnetic reconnection, which is demonstrated by the presence of TRINNI return flows (thick black curve). Furthermore, the reconnection points should retreat to further down-tail as the nightside distorted TPA grows, as shown with green stars in the duskside (surrounded by a blue box). Nowada et al. (2020) showed that reconnection-associated upward field-aligned currents (FACs), plausibly triggered by nightside reconnection as shown with purple solid arrows, can be a "source" of the "J"-shaped TPA (magenta shading). In this case, however, conclusive signatures on the existence of upward FACs around the TPA cannot be shown because, unfortunately, the magnetic field data obtained by low-altitude orbiters were absent during the interval of interest. Instead, we are able to show some indication of the TPA-associated FAC flowing sense, which is provided by equivalent ionospheric current (EIC), estimated based on the geomagnetic field data from the SuperMAG ground observatory network (Gjerloev, 2012).

In Figure 7, EIC distributions on 9:26 UT (panel a), 10:00 UT (panel b), and 10:10 UT (panel c), projected onto the IMAGE FUV-WIC imager data in geomagnetic coordinates, are shown to estimate the orientation and scale of the FAC system around the growing "J"-shaped TPA. The EIC vectors (red bars) are derived by rotating the horizontal magnetic field components (local magnetic north – south and east – west components) 90 degrees clockwise using the same calculation techniques, proposed by Glassmeier et al. (1989), Moretto et al. (1997), Motoba et al. (2003), and references therein. The major trends of the EIC vectors in close proximity to the regions of growth of the "J"-shaped TPA (magenta thick and curved arrows) exhibit a significant counter-clockwise rotation, implying that upward FACs might be generated around the "J"-shaped TPA. Significant distortion-aligned duskward plasma flows from around the "J"-shaped TPA growth point (orange curved arrow in Figure 6) can be explained within a framework of Dungey-cycle-driven plasma flow patterns in the polar region. The nightside ionospheric flow patterns during the "J"-shaped TPA, as seen in Figures 3, 4 and 5, clearly have dawn-dusk asymmetry, suggesting that the IMF-$B_y$ component influenced the nightside magnetosphere, that is, the nightside plasma sheet deformation and magnetic field line twisting in the magnetotail were caused by the IMF-$B_y$ penetration (e.g., Cowley, 1981, 1994, Milan et al. 2005, Fear and Milan, 2012a). Nowada et al. (2020) demonstrated that the nightside distorted TPAs also grow to the dayside under dawn-dusk asymmetric ionospheric flow patterns. In this study, we are able to

reveal the details of the ionospheric flow patterns that cause the nightside distorted part of the TPA using ground-based radar observations.

The scenario for the formation of nightside distortion of a TPA and possible whole TPA growth model, illustrated in Figure 6 and Nowada et al. (2020), can be applied to "J"- and "L"-shaped TPAs formed during northward IMF intervals. Table 1 shows the event number of the nightside distorted TPA, categorized by three types of the IMF-$B_z$ polarity based on the 17 nightside distorted TPA events, which were selected from the IMAGE FUV-WIC observations from 2000 to 2005 and include the 9 TPA events used in Nowada et al. (2020) (see Table S1). In most TPA cases during the northward IMF intervals, TRINNI flow signatures were detected with the SuperDARN radar arrays, suggesting that magnetotail reconnection occurrences and the explanations with reconnection-based nightside distorted TPA formation process can be expected. In contrary, the nightside distorted TPA under purely southward conditions is only two events. In the case discussed in this study, we show that the TPA nightside distortion formation, and explain a possible whole "J"-shaped TPA growth, adopting a nightside distorted TPA formation model under northward IMF conditions (Nowada et al. 2020) while neither satellite nor SuperDARN radar profiles can be obtained in another case. Therefore, it is required to discuss much enough space- and ground-observation data in considering general nightside distorted TPA formation processes under southward IMF conditions.

|  | IMF-$B_z > 0$ | IMF-$B_z < 0$ | From IMF-$B_z > 0$ To IMF-$B_z < 0$ |
|---|---|---|---|
| "J"-shaped TPA | 8 | 2 | 1 |
| "L"-shaped TPA | 5 | 0 | 1 |
| Total | 13 | 2 | 2 |

**Table 1: The event number of the nightside distorted TPA, which was categorized by three types of the IMF-Bz polarity (northward IMF: IMF-$B_z > 0$; southward IMF: IMF-$B_z < 0$; turning from northward to southward IMF: from IMF-$B_z > 0$ to IMF-$B_z < 0$) is shown. This table is made based on the 17 nightside distorted TPA events, which were selected from the IMAGE FUV-WIC observations from 2000 to 2005 (see Table S1)**

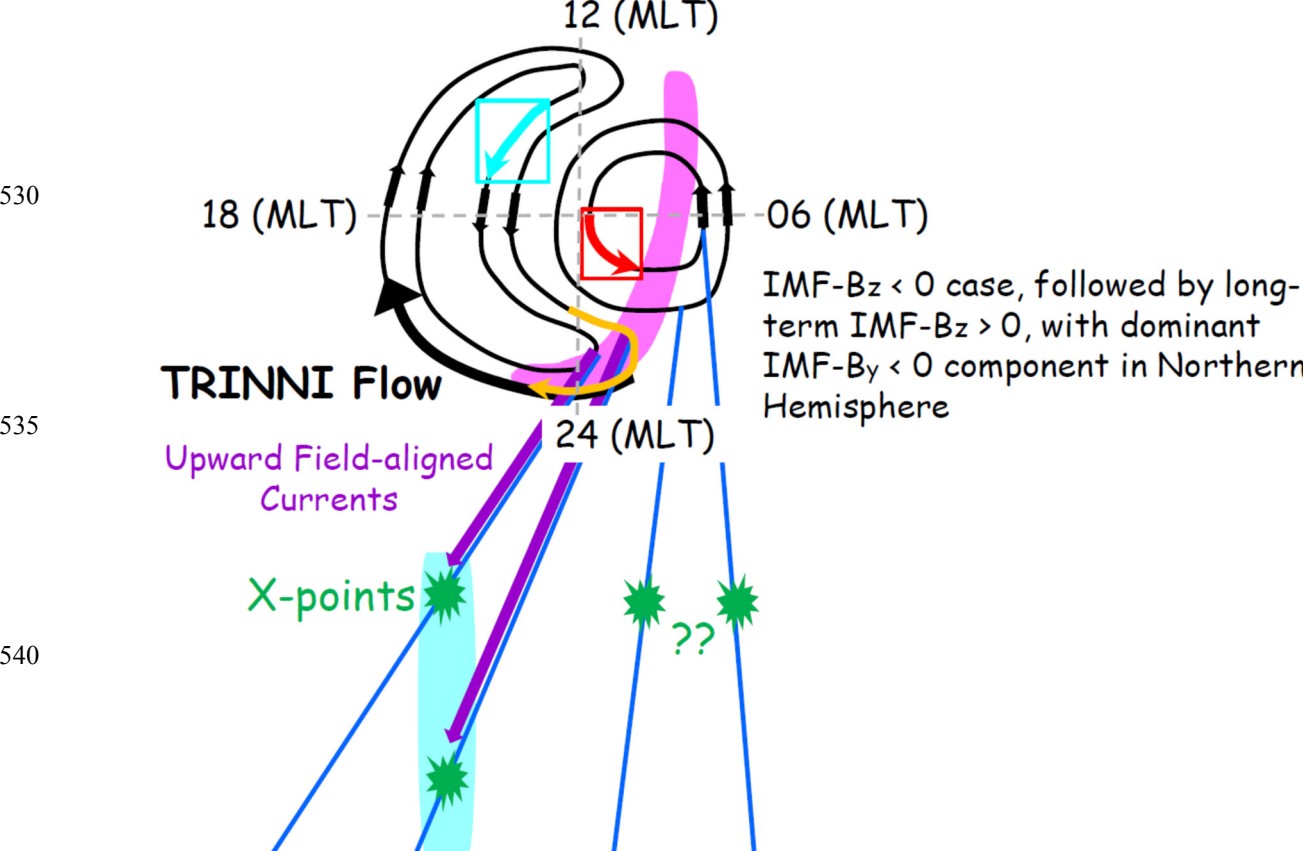

12 (MLT)

18 (MLT)

06 (MLT)

IMF-Bz < 0 case, followed by long-term IMF-Bz > 0, with dominant IMF-By < 0 component in Northern Hemisphere

TRINNI Flow

24 (MLT)

Upward Field-aligned Currents

X-points

??

**Figure 6: A schematic illustration of the global ionospheric convection flow patterns in the northern hemisphere driven by dayside and nightside magnetic reconnections during the growth interval of the "J"-shaped TPA (magenta) is shown. This illustration is modified Figure 3 of Grocott et al. (2005). These ionospheric flow patterns are expected to be seen when the IMF-B$_z$ are southward with dominant dawnward component. The ionospheric flows, highlighted by thick cyan and red curved arrows and surrounded with cyan and red squares, are corresponding to those highlighted with the same colored squares, as shown in Figure 5(b). The TRINNI return flows and plasma flows out of the TPA growth point, which may lead to the formations of the observed "J"-shaped TPA and its nightside end distortion, are shown with black thick and orange curved arrows, respectively. The reconnection-generated upward field-aligned currents (FACs) as a source of nightside distorted TPA are shown with purple solid arrows. The blue solid lines indicate magnetospheric closed field lines. The reconnection points, retreating as the TPA grows to the dayside, are shown with green stars and their retreat line is highlighted with a blue box. The day- night and dawn-dusk meridian lines are shown with gray broken lines.**

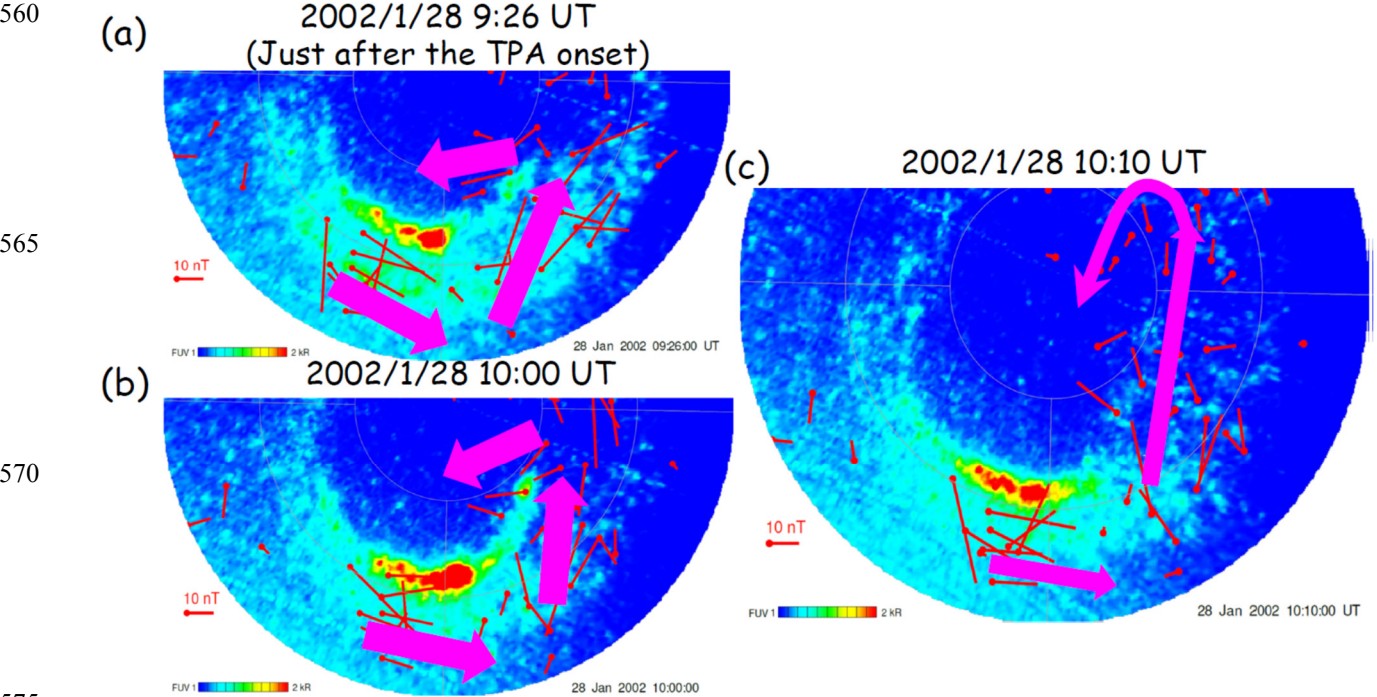

**Figure 7: Equivalent ionospheric current (EIC) distributions on 9:26 UT (panel a), 10:00 UT (panel b), and 10:10 UT (panel c), projected onto the IMAGE FUV-WIC data in geomagnetic coordinates, are shown. The EIC vectors (red bars) are derived by rotating the horizontal magnetic field components (local magnetic north – south and east – west components) 90 degrees clockwise. The geomagnetic field was measured at the ground magnetic observatories from the SuperMAG ground observatory network (Gjerloev, 2012). Each panel is oriented such that the right, bottom and left sides are corresponding to dawn (6h), midnight (24h), and dusk (18h) in MLT, respectively. The major directional trends of the EIC vectors in close proximity to the regions of growth of the "J"-shaped TPA are shown with magenta thick and curved arrows. The white circles show the MLat values from 60 degrees to 80 degrees as they go inward. The color codes are assigned according to unit of Rayleigh.**

### 5.2 Significant Differences between Nightside Distorted TPA, Bending Arcs, and Double Oval

Given the variety of different TPA morphologies discussed previously in the literature, we here briefly outline the main differences between the distorted arc discussed in this paper, and bending arcs and double oval auroral forms.

### 5.2.1 Difference from Bending Arcs

The most significant difference between nightside distorted TPAs and bending arcs is whether or not nightside magnetic reconnection is closely related with its formation (e.g., Kullen et al. 2015). During our "J"-shaped TPA interval, TRINNI

return flows were clearly observed in the poleward edge of the nightside main auroral oval (near the growth point of the "J"-shaped TPA), suggesting that nightside reconnection persisted even during the TPA development and, according to our proposed mechanism, plays an essential role in the "J"-shaped TPA formation. Bending arcs, on the other hand, tend to occur in association with dayside magnetic reconnection (Kullen et al., 2015) and thus have a different formation process to the nightside distorted TPA presented here. In this study, the "J"-shaped TPA was observed during a southward IMF interval, and ionospheric plasma flow patterns indicating the occurrence of low-latitude dayside magnetic reconnection were also observed. These IMF conditions and associated ionospheric flow profiles are actually consistent with those found when bending arcs are likely to be formed (Kullen et al., 2015, Carter et al., 2015). However, in this case, even in the presence of dayside magnetic reconnection, which drove Dungey-cycle-driven plasma flow patterns, this TPA displayed no characteristics of a bending arc, with its growth toward the dayside being rather straightforward. Because bending arcs develop toward the dawn or dusk sector in a region more poleward of the dayside magnetic reconnection line (merging gap) (Carter et al., 2015), the development profile of the "J"-shaped TPA is also different from that of bending arc. Therefore, we can be certain that the nightside distorted TPA discussed in our study is quite distinct from "bending arcs".

### 5.2.2 Difference from Double Auroral Oval

In association with our "J"-shaped TPA westward (duskside) flow, both equatorward and poleward of the TPA nightside distortion was observed in the SuperDARN data, seen in the plasma flow patterns after the TPA onset (several panels of Figures 2, 3, 4 and 5). These suggest that upward currents (FACs) should flow in the regions equatorward and poleward of the nightside distortion of the TPA, although a detailed FAC profile cannot be estimated based on the EIC vectors poleward of the TPA nightside distortions because of an absence of sufficient geomagnetic field data (see Figure 7). In the region just poleward of the main auroral oval, including the distorted part of TPA (panels a and b in Figure 7), we can suggest the presence of upward FACs around the distortion at the TPA nightside end. These FAC profiles around the nightside distortion of the TPA are inconsistent with the double auroral oval structures elucidated by Ohtani et al (2012) (see their Figure 11). Therefore, the nightside distortion of the "J"-shaped TPA is also independent of the double auroral oval.

### 6. Conclusions

Nightside magnetic reconnection and associated FACs are integral processes to form nightside distorted TPAs, such as the "J"-shaped TPA presented in this study. In particular, a migration of the equatorward plasma flows, which rotated to align with the main auroral oval, at the point where the TPA starts to protrude into the polar cap toward the dayside, plays a significant role in the formation of the distorted nightside end of the TPA. These plasma flows may be interpreted within a framework of the dawn-dusk asymmetric polar cap plasma flow patterns, produced by ongoing Dungey-cycle activity in the presence of a dominant IMF-$B_y$ component. From the global ionospheric plasma flow patterns determined from SuperDARN radar observations, we can conclude that nightside distorted TPAs are formed by a juxtaposition of localized flow stagnation (as required for "regular" TPAs) in the presence of ongoing "TRINNI"-type tail-reconnection-driven-flows consistent with

the distortion of the TPA nightside end. It may also be the case that this process is facilitated by a southward IMF, and associated ongoing dayside reconnection, that is required to "feed" the TRINNI flows. As such, nightside distorted TPAs, including the "J"-shaped TPA in this study, are quite different from both "double oval" and "bending arcs" in terms of their formation process.

In the near future, the SMILE (Solar wind Magnetosphere Ionosphere Link Explorer) and STORM (Solar-Terrestrial Observer for the Response of the Magnetosphere) satellites, which include auroral UVI imagers with higher spatial and temporal resolutions than those on the Polar and IMAGE missions, will be launched. If the UVI auroral imager data can safely be acquired after a successful launch of these new satellites, we can expect to collect more nightside distorted TPA events, and to study more closely the detailed features and formation mechanism of these "J" and "L"-shaped TPAs.

**Data Availability:** All SuperDARN radar data are processed by the software of fitacf v1.2 and make_grid v1.14.er and can be obtained from https://www.bas.ac.uk/project/superdarn. IMAGE FUV-WIC data were accessed from http://image.gsfc.nasa.gov. Solar wind OMNI (ACE MFI and SWE) data were obtained from Coordinated Data Analysis Web (https://cdaweb.sci.gsfc.nasa.gov/index.html/), provided by GSFC/NASA. The ground magnetometer data were obtained from the website of the SuperMAG ground observatory network (https://supermag.jhuapl.edu/).

**Author Contribution:** MN and AG wrote the draft of the manuscript and performed the data analysis. QQS is the PI of the main big research project in which MN is taking part.

**Competing Interests:** The author declares that they have no conflict of interest.

**Acknowledgments:** M.N. greatly thanks Robert C. Fear, Alexander William Degeling, An-Min Tian, and Jong-Sun Park for fruitful and promotive discussions, with particular thanks to Robert C. Fear for carefully reading the initial version of this manuscript. Also, he thanks Benoît Hubert for helping to process the IMAGE FUV-WIC data. We thank the PIs of the SuperDARN radars for provision of the ionosphere flow data. SuperDARN is funded by the research agencies of Australia, China, Canada, France, Italy, Japan, South Africa, the U. K. and the U. S. For the ground magnetometer data, we gratefully acknowledge: Intermagnet; USGS, Jeffrey J. Love; CARISMA, PI Ian Mann; CANMOS, Geomagnetism Unit of the Geological Survey of Canada; The S-RAMP Database, PI K. Yumoto and Dr. K. Shiokawa; The SPIDR database; AARI, PI Oleg Troshichev; The MACCS program, PI M. Engebretson; GIMA; MEASURE, UCLA IGPP and Florida Institute of Technology; SAMBA, PI Eftyhia Zesta; 210 Chain, PI K. Yumoto; SAMNET, PI Farideh Honary; The IMAGE magnetometer network, PI L. Juusola; AUTUMN, PI Martin Connors; DTU Space, PI Anna Willer; South Pole and McMurdo Magnetometer, PI's Louis J. Lanzarotti and Alan T. Weatherwax; ICESTAR; RAPIDMAG; British Artarctic Survey; McMac, PI Dr. Peter Chi; BGS, PI Dr. Susan Macmillan; Pushkov Institute of Terrestrial Magnetism, Ionosphere and Radio Wave Propagation (IZMIRAN); GFZ, PI Dr. Juergen Matzka; MFGI, PI B. Heilig; IGFPAS, PI J. Reda;

University of L'Aquila, PI M. Vellante; BCMT, V. Lesur and A. Chambodut; Data obtained in cooperation with Geoscience Australia, PI Marina Costelloe; AALPIP, co-PIs Bob Clauer and Michael Hartinger; SuperMAG, PI Jesper W. Gjerloev; Sodankylä Geophysical Observatory, PI Tero Raita; Polar Geophysical Institute, Alexander Yahnin and Yarolav Sakharov; Geological Survey of Sweden, Gerhard Schwartz; Swedish Institute of Space Physics, Mastoshi Yamauchi; UiT the Arctic University of Norway, Magnar G. Johnsen; Finish Meteorological Institute, PI Kirsti Kauristie.

**Financial Support:** This work and MN are supported by grant of the National Natural Science Foundation of China (NSFC 42074194). QQS is supported by NSFC 41731068, 41961130382 and 41974189, and also supported from International Space Science Institute, Beijing (ISSI-BJ). AG is supported by STFC grant (ST/R000816/1) and NERC grants (NE/P001556/1 and NE/T000937/1).

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
