# Peer review of "Ionospheric Plasma Flows Associated with the Formation of the Distorted Nightside End of A Transpolar Arc"

_Annales Geophysicae, 2021_

## Author Comment (AC1)

**Reply to RC 1 (ANGEO-2021-69)**

We appreciate very much the referee's kind and careful comments and useful suggestions to our manuscript entitled "Ionospheric plasma flows associated with the formation of the distorted nightside end of a transpolar arc (ANGEO-2021-69)".

We have revised the manuscript in accordance with each of the comments and suggestions from Referee1.

The referee's comments are written in Red and our reply is written in Blue.

Specific Comments:

1. The paper contains an extremely detailed analysis of a single "J"-shaped TPA observed on January 28, 2002, however it is not clear how generalizable the specific conclusions are to other distorted TPAs. The paper could be improved with a discussion of how common "J"-shaped TPAs are and if this description of formation should be expected to apply to most "J"-shaped TPAs. If there is insufficient data to show several examples of similar events, please state that directly.

Thank you very much for your suggestions. We added one paragraph and one table as shown from Ls. 493 to 514 to show that this event discussed is rare case, compared with the other nightside distorted TPAs (not only "J"- but also "L"-shaped TPAs) in terms of "phenomenology" and "difficulty of data collection". The formation processes of the TPA nightside distortion and possible whole "J"-shaped TPA are discussed based on the nightside distorted TPA formation model proposed by Nowada et al. (2020), which was constructed by the nightside distorted TPAs under northward IMF conditions. In this study, we pointed out that main concept of the Nowada's model can also be applied to the "J"-shaped TPA under southward IMF conditions.

"The scenario for the formation of nightside distortion of a TPA and possible TPA growth model, illustrated in Figure 6 and Nowada et al. (2020), can be applied to "J"- and "L"-shaped TPAs formed during northward IMF intervals. Table 1 shows the event number of the nightside distorted TPA, categorized by three types of the IMF-Bz polarity based on the 17 nightside distorted TPA events, which were selected from the IMAGE FUV-WIC observations from 2000 to 2005 and include the 9 TPA events used in Nowada et al. (2020) (see Table S1). In most TPA cases during the northward IMF intervals, TRINNI flow signatures were detected with the SuperDARN radar arrays, suggesting that magnetotail reconnection occurrences and the explanations with reconnection-based nightside distorted TPA formation process can be expected. In contrary, the nightside distorted TPA under purely southward conditions is only two events. In the case discussed in this study, we show that the TPA nightside distortion formation, and explain a possible "J"-shaped TPA growth, adopting a nightside distorted TPA formation model under northward IMF conditions (Nowada et al. 2020) while neither satellite nor SuperDARN radar profiles can be obtained in another case. Therefore, it is required to discuss much enough space- and ground-observation data in considering general nightside distorted TPA formation processes under southward IMF conditions."

2. Due to the importance of SuperDARN convection maps in this paper, a more detailed description of their construction and features are warranted. Specifically, consider discussing some of the results from this (very) recent paper and whether they would influence the accuracy of the convection maps used in the analysis.

Walach, M.-T., Grocott, A., Staples, F., & Thomas, E. G. (2022). Super Dual Auroral Radar Network Expansion and its Influence on the Derived Ionospheric Convection Pattern. Journal of Geophysical Research: Space Physics, 127, e2021JA029559. https://doi.org/10.1029/2021JA029559.

Thank you very much for your suggestions. We added a detailed description of the steps taken to construct the convection maps, and a discussion of the relevant points from the suggested paper in the manuscript. Details were described in Section 2.

3. Figure S1 is referenced several times in the manuscript, and Section 5.2.2 is difficult to understand without it. Please consider adding Figure S1 to the main manuscript to make it easily accessible to readers.

Thank you very much for your suggestions. We added Figure S1 and associated descriptions as Figure 7 in the main manuscript along your suggestion. From figure 7, it can be easy to know the FAC scale and orientation sense from the EIC (equivalent ionospheric currents) distributions during the growth of "J"-shaped TPA discussed in this study.

Technical Comments:
Line 49: no -> not
Corrected. (L. 50)

Line 104: Remove comma – "much as possible using the methods described in Nowada et al. (2020)"
Removed. (L. 106)

Figure 1/2 caption: Rephrase "The circles drawn from outer- to inner sides in each panel…" to "The concentric circles in each panel show MLAT at 60, 70, and 80 degrees." This decreases the ambiguity of what MLAT the middle circle represents.
Thank you very much. We replaced the old sentence with a sentence where you suggested. (Ls.177 – 178, and Ls. 217 – 218)

Figure 2 caption: Please make sure the clock angle equation uses the correct super and sub-scripts. Throughout the paper (particularly between body text and captions) there is some inconsistency as to whether the IMF components x, y, and z are subscripted or not (i.e. IMF-Bz vs IMF-Bz). Either is fine, but please be consistent throughout the paper.
Thank you very much. We revised all subscripts used in this paper, such as "IMF-$B_z$ and IMF-$B_y$". On the equation of clock angle, we revised the formula, such as "arctan(IMF-$B_y$/IMF-$B_z$)"

Line 190: Change "A negative bay in the AL index with a peak around -150 nT occurred from 7:30 to 8:10" to "The dip in the AL index down to -150 nT between 7:30 and 8:10 UT suggests …"
Thank you very much. We revised this sentence along your suggestion. (L. 222)

Line 212 and elsewhere: Please discuss flows in units of m/s to match the plot units.
Along your suggestion, we replaced all plasma flow units as used in this paper [km/s] with [m/s] to match

the flow units seen in the SuperDARN plots. Please see Ls. 37 – 38, 244, 279, 386, and 418.

Line 423: "Super DARN" -> "SuperDARN"
  Corrected. (L. 457)

Figure 6 caption: Describing the TPA in the figure as "thin magenta" is somewhat confusing as the region
  in the figure is quite thick. Considering simply identifying it as "magenta".
  Thank you very much. We changed a terminology of "thin magenta" in the caption of Figure 6 to simply
  "magenta".

---

## Author Comment (AC2)

**Reply to RC 2 (ANGEO-2021-69)**

We appreciate very much the referee's kind and useful suggestion to our manuscript entitled "Ionospheric plasma flows associated with the formation of the distorted nightside end of a transpolar arc (ANGEO-2021-69)".

We have revised the manuscript in accordance with the suggestion from Referee2.

The referee's comment is written in Red and our reply is written in Blue.

L9 "flow patterns from 28th January 2002" should read "flow patterns occurring on 28 January 2002"

Thank you very much for your suggestion. We revised this sentence as you pointed out.

We also revised the relevant points of your suggestion. Please see Ls. 173, 184, and 208.